# Improving probabilistic infectious disease forecasting through coherence

**Graham Casey Gibson** [1,2]*, **Kelly R. Moran** [1,3], **Nicholas G. Reich** [2], **Dave Osthus** [1]

**1** Statistical Sciences Group, Los Alamos National Laboratory, Los Alamos, New Mexico, United States of America, **2** Department of Biostatistics and Epidemiology, University of Massachusetts-Amherst, Amherst, Massachusetts, United States of America, **3** Department of Statistical Science, Duke University, Durham, North Carolina, United States of America

* gcgibson@umass.edu

**Data Availability Statement:** FluSight challenge submission files are available at https://github.com/cdcepi/FluSight-forecasts. Census data are available at

## Abstract

With an estimated $10.4 billion in medical costs and 31.4 million outpatient visits each year, influenza poses a serious burden of disease in the United States. To provide insights and advance warning into the spread of influenza, the U.S. Centers for Disease Control and Prevention (CDC) runs a challenge for forecasting weighted influenza-like illness (wILI) at the national and regional level. Many models produce independent forecasts for each geographical unit, ignoring the constraint that the national wILI is a weighted sum of regional wILI, where the weights correspond to the population size of the region. We propose a novel algorithm that transforms a set of independent forecast distributions to obey this constraint, which we refer to as probabilistically coherent. Enforcing probabilistic coherence led to an increase in forecast skill for 79% of the models we tested over multiple flu seasons, highlighting the importance of respecting the forecasting system's geographical hierarchy.

## Author summary

Seasonal influenza causes a significant public health burden nationwide. Accurate influenza forecasting may help public health officials allocate resources and plan responses to emerging outbreaks. The U.S. Centers for Disease Control and Prevention (CDC) reports influenza data at multiple geographical units, including regionally and nationally, where the national data are by construction a weighted sum of the regional data. In an effort to improve influenza forecast accuracy across all models submitted to the CDC's annual flu forecasting challenge, we examined the effect of imposing this geographical constraint on the set of independent forecasts, made publicly available by the CDC. We developed a novel method to transform forecast densities to obey the geographical constraint that respects the correlation structure between geographical units. This method showed consistent improvement across 79% of models and that held when stratified by targets and test seasons. Our method can be applied to other forecasting systems both within and outside an infectious disease context that have a geographical hierarchy.

https://www2.census.gov/programs-surveys/popest/datasets/2010-2018/national/totals/nst-est2018-alldata.csv.

**Funding:** This work was funded by the Department of Energy at Los Alamos National Laboratory under contract 89233218CNA000001 through the Laboratory-Directed Research and Development Program, specifically LANL LDRD grant 20190546ECR. This work has also been supported by the National Institutes of General Medical Sciences (R35GM119582) and by the US Centers for Disease Control and Prevention (1U01IP001122). The funders had no role in study design, data collection and analysis, decision to publish, or preparation of the manuscript.

**Competing interests:** The authors have declared that no competing interests exist.

## Introduction

Seasonal influenza is a persistent and serious contributor to global morbidity and mortality, hospitalizing over half a million people in the world every year [1]. The United States alone reported approximately 80,000 influenza related mortalities in the 2017/2018 influenza season, with most serious consequences for vulnerable populations such as children or the elderly [2].

As part of a larger forecasting initiative, the U.S. Centers for Disease Control and Prevention (CDC) hosts an annual influenza forecasting challenge called the FluSight challenge open to the public [3, 4]. As part of this challenge, forecasters supply probabilistic forecasts for short-term and seasonal targets at both the national and regional levels The target of interest is weighted influenza-like illness (wILI), which measures the proportion of outpatient doctor visits at reporting health care facilities where the patient had an influenza-like illness (ILI), weighted by state population. At the national level, wILI can be directly computed using a sum of state population weighted ILI or it can be equivalently computed using regional population weighted wILI. The CDC estimates ILI as the ratio of patients presenting with a fever equal to or above 100˚ Fahrenheit, a cough or sore throat and no other known cause. over the total number of patients presenting at health care providers [5].

Participants in the FluSight challenge have harnessed a variety of models and methods to forecast the targets under consideration, which include both short-term forecasts and seasonal targets. These efforts have included time series models [6], mechanistic disease transmission models [7, 8], and machine learning techniques [9–12]. Teams have also incorporated external data, such as internet search queries or point of care data, to improve forecasts [13–16]. FluSight challenge participation has grown in popularity since the inaugural challenge in 2013, with twenty-four teams submitting forecasts from thirty-three models for the 2018/2019 season [17]. Model submission files from the past FluSight challenges are publicly available [17], providing the opportunity for retrospective analysis and the potential for improved forecasting.

Multiple procedures have been proposed in the literature to transform independently generated incoherent forecasts into coherent forecasts, also called forecast reconciliation [18, 19]. We follow the common definition of "coherent", where a set of forecasts to be coherent if they respect the hierarchical data generating process [18]. Projection matrix forecasting is a popular coherence forecasting approach [18]. This approach uses a matrix projection of the original set of forecasts onto a subspace that respects the known hierarchical relationship of the forecasting system. This approach uses forecasts for all levels of the hierarchy and does not discard any information as opposed to say a bottom-up approach, where the national forecast is ignored and the estimate is simply a linear combination of regional estimates. However, the demonstrated benefits of coherence in the point prediction setting do not necessarily translate to the probabilistic forecasting realm [19]. Previous work has focused on applying probabilistic coherence constraints after estimating regional weights based on historical data. While this is appealing from an optimization perspective, it limits new participants in the FluSight challenge, as well as other applications with limited forecast accuracy data, such as in emerging epidemics and novel surveillance networks. We propose a novel algorithm that transforms a set of independent forecast densities to satisfy the probabilistic coherence property without historical forecast accuracy data and demonstrate that the resulting collection of forecast densities has a consistently higher forecast skill when broken down by season and target.

## US ILI surveillance data

For the FluSight challenge, the CDC provides wILI data at both the national level and broken down into 10 Health and Human Services (HHS) regions, mostly organized by geographic proximity. The data are reported on a weekly basis and extend from 1997 to the present.

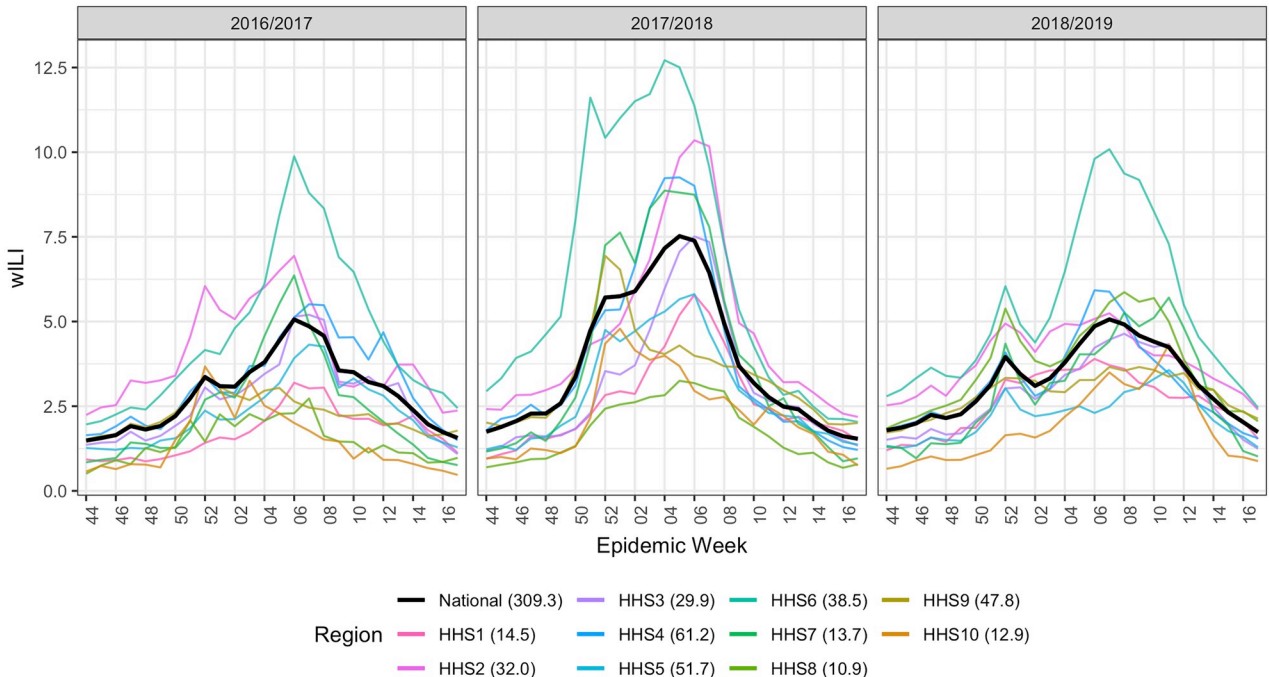

**Fig 1. Data example for the three test seasons under consideration (2016/2017, 2017/2018, 2018/2019) season for all 10 Health and Human Services (HHS) regions and the national level.** At any given epiweek, the national wILI (black) is a weighted sum of regional wILI, where the weights correspond to the population size of the region. We can see that wILI is highly seasonal and varies heavily by region. Region population sizes (in millions) are given next to the region in the legend.

Example data for the 2016/2017, 2017/2018, and 2018/2019 seasons are shown in Fig 1. As noted in the Fig 1, wILI varies by region but maintains a relatively consistent winter peak. The CDC reports the wILI data using epidemic weeks, called epiweeks, instead of calendar weeks [20]. This allows for consistent week numbering across multiple seasons. Epiweek 40 is usually the first week of October, the start of the flu season, and epiweek 20 usually falls in May, marking the end of the season.

## Pointwise forecast coherence

The partitioning of national data into HHS regions facilitates geographically localized forecasts, augmenting their usability to local public health officials. A consequence of this partitioning, however, is the creation of a hierarchical structure in the forecasting system. Namely, national wILI data is a linear combination of HHS regional wILI data. Region population sizes (in millions) are given next to the region in the legend of Fig 1 as reported by the 2010 U.S. Census [21].

We notate the true wILI value as $y_{r,s,w} \in [0, 100]$, a percentage for region $r$ in flu season $s$ corresponding to epiweek $w$. Throughout the paper, $r = 11$ corresponds to the nation, while $r = 1, 2, \ldots, 10$ corresponds to HHS region $r$. Let $\alpha_r \in [0, 1]$ be a weight corresponding to HHS region $r$, proportional to the population of HHS region $r$, such that $\sum_{r=1}^{10} \alpha_r = 1$. The hierarchical nature of the national/regional partitioning of forecasts for any season and epiweek is equivalent to the following constraint:

$$y_{11,s,w} = \sum_{r=1}^{10} \alpha_r y_{r,s,w}. \tag{1}$$

For convenience, define the collection of point forecasts for all regions for season $s$ and epi-week $w$ as

$$\tilde{\boldsymbol{y}}_{s,w}^T \quad = [\tilde{y}_{1,s,w}, \tilde{y}_{2,s,w}, \ldots, \tilde{y}_{11,s,w}]. \tag{2}$$

We say that the forecast $\tilde{\boldsymbol{y}}_{s,w}$ is *coherent* if

$$\tilde{y}_{11,s,w} \quad = \sum_{r=1}^{10} \alpha_r \tilde{y}_{r,s,w}, \tag{3}$$

and $\tilde{\boldsymbol{y}}_{s,w}$ is *incoherent* if

$$\tilde{y}_{11,s,w} \quad \neq \sum_{r=1}^{10} \alpha_r \tilde{y}_{r,s,w}. \tag{4}$$

Though the existence of the hierarchical structure in the wILI forecasting system is know, many FluSight challenge forecasts are made independently. Forecasting at geographic regions independently provides the forecaster more flexibility to cater models to specific regions or avoid modeling correlation between regions explicitly, but leaves the resulting forecasts vulnerable to incoherence as the true coherent data generating process is not respected. In this paper, we use $\tilde{\boldsymbol{y}}$ to represent independently generated and (likely) incoherent forecasts and $\hat{\boldsymbol{y}}$ to represent coherent forecasts.

For an independently generated set of forecasts $\tilde{\boldsymbol{y}}$, the corresponding coherent projection matrix forecast $\hat{\boldsymbol{y}}$ is

$$\hat{\boldsymbol{y}} = \boldsymbol{X}(\boldsymbol{X}^T \boldsymbol{V} \boldsymbol{X})^{-1} \boldsymbol{X}^T \boldsymbol{V} \tilde{\boldsymbol{y}} = \boldsymbol{P} \tilde{\boldsymbol{y}}, \tag{5}$$

where $\boldsymbol{X}$ is a design matrix corresponding to the hierarchical relationship of the data generating process and $\boldsymbol{V}$ is a weight matrix. Specifically for the FluSight challenge,

$$\boldsymbol{X}_{11 \times 10} = \begin{pmatrix} \boldsymbol{I}_{10 \times 10} \\ \boldsymbol{\alpha}_{1 \times 10} \end{pmatrix} \tag{6}$$

$$\boldsymbol{\alpha} = [\alpha_1, \alpha_2, \ldots, \alpha_{10}] \tag{7}$$

where $\alpha_r$ is the weight for the $r^{th}$ region. We can think of this as a linear regression model with a design matrix $\boldsymbol{X}$ that enforces coherence. Therefore, any projection into the column space of $\boldsymbol{X}$, must preserve coherence.

A special case of projection matrix forecasting is when the ordinary least squares (OLS) projection matrix is used, produced by setting $\boldsymbol{V}$ equal to the identity matrix $\boldsymbol{I}$ in Eq 5. This special case has the property that the resulting coherent point forecast $\hat{\boldsymbol{y}}$ has mean squared error (MSE) no worse than the independently generated forecast $\tilde{\boldsymbol{y}}$ [19]. That is, when $\boldsymbol{V} = \boldsymbol{I}$ in Eq 5,

$$||\tilde{\boldsymbol{y}} - \boldsymbol{y}||_2 \quad \geq ||\hat{\boldsymbol{y}} - \boldsymbol{y}||_2 \tag{8}$$

for any $\boldsymbol{y}$ and $\tilde{\boldsymbol{y}}$ (See S1 Appendix for proof).

For illustration and clarity, consider an example with two low level regions (HHS1 and HHS2) and one top level region (Nation). This example is illustrated in Fig 2.

Assume

$$y_{\text{Nat}} \quad = \alpha_1 \ y_{\text{HHS1}} + \alpha_2 \ y_{\text{HHS2}}, \tag{9}$$

where $\alpha_1 = \alpha_2 = 0.5$. Let the true values be

$$\boldsymbol{y}^T = [y_{\text{HHS1}}, y_{\text{HHS2}}, y_{\text{Nat}}] = [1, 1, 1], \tag{10}$$

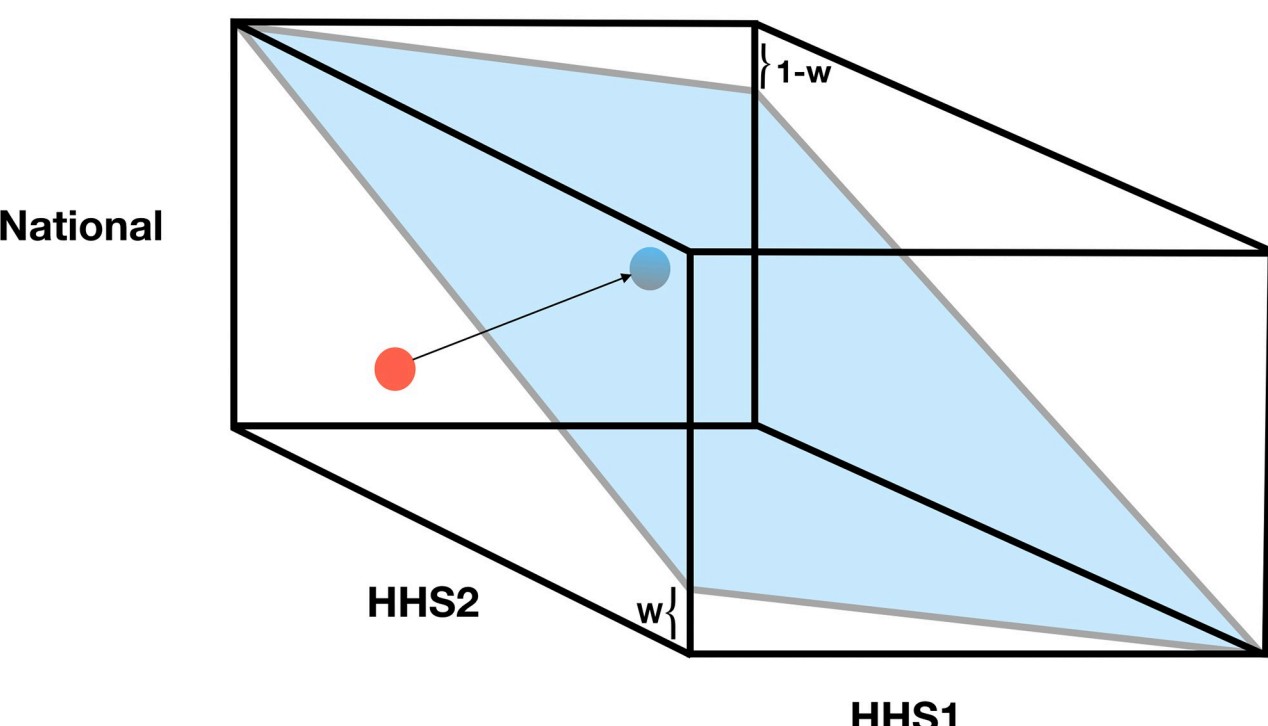

**Fig 2. Mock example of independent forecasts (red) and projected forecasts (blue) for three regions, National, HHS1 and HHS2.** Both the blue and the red point represent a triple of ILI forecast values for each region. Independent forecasts are projected onto the space satisfying the constraint of regional level forecasts summing to national level. The blue plane represents the set of points that satisfy the coherence constraint, namely that the weighted combination of region-level forecasts equals the National level forecast. Different projection matrices are able to map the red point to the blue point at different locations on the blue plane.

and assume the independently generated forecasts are

$$\tilde{\boldsymbol{y}}^T = [\tilde{y}_{\text{HHS1}}, \tilde{y}_{\text{HHS2}}, \tilde{y}_{\text{Nat}}] = [1/2, 1/2, 1]. \tag{11}$$

Notice that the independently generated forecasts are incoherent, as

$$\tilde{y}_{\text{Nat}} = 1 \neq 1/2 = \alpha_1\,\tilde{y}_{\text{HHS1}} + \alpha_2\,\tilde{y}_{\text{HHS2}}. \tag{12}$$

The MSE for $\tilde{\boldsymbol{y}}$ is

$$||\tilde{\boldsymbol{y}} - \boldsymbol{y}||_2 = \frac{1}{3}\left((1/2 - 1)^2 + (1/2 - 1)^2 + (1 - 1)^2\right) = 1/6. \tag{13}$$

The OLS projection matrix forecast is $\hat{\boldsymbol{y}} = [2/3, 2/3, 2/3]$, computed as

$$\hat{\boldsymbol{y}} = \boldsymbol{X}(\boldsymbol{X}^T\boldsymbol{X})^{-1}\boldsymbol{X}^T\tilde{\boldsymbol{y}}, \tag{14}$$

where

$$\boldsymbol{X} = \begin{pmatrix} 1 & 0 \\ 0 & 1 \\ 1/2 & 1/2 \end{pmatrix}.$$

The projection matrix forecast $\hat{y}$ is, by construction, coherent. The effect of the projection matrix forecast is a reduction in MSE over $\tilde{y}$, where the MSE for $\hat{y}$ is

$$||\hat{\boldsymbol{y}} - \boldsymbol{y}||_2 = \frac{1}{3}\left((2/3-1)^2 + (2/3-1)^2 + (2/3-1)^2\right) = 1/9. \tag{15}$$

Note the MSE for $\hat{y}_{\text{HHS1}}$ and $\hat{y}_{\text{HHS2}}$ improved relative to $\tilde{y}_{\text{HHS1}}$ and $\tilde{y}_{\text{HHS2}}$, respectively, while the MSE for $\hat{y}_{\text{Nat}}$ got worse relative to $\tilde{y}_{\text{Nat}}$, resulting in an overall, but not uniform, improvement in MSE for $\hat{y}$ relative to $\tilde{y}$. The projection matrix forecast is a useful tool for transforming independently generated, incoherent forecasts into coherent forecasts. When $V = I$ in Eq 5, the MSE of the resulting $\hat{y}$ is guaranteed to be no greater than the MSE of $\tilde{y}$. When $V \neq I$ in Eq 5, the resulting forecast is still coherent, but no such guarantee of MSE improvement exists.

## The FluSight forecast coherence dilemma

Unlike the situations demonstrated in the point forecasting literature, forecasts for the FluSight challenge are required to be probabilistic, not point estimates, and the probabilistic forecasts are evaluated using a multi-bin scoring rule, not MSE. In practice, probabilistic forecasts are generated as a collection of $n$ forecast samples for $y_{r,s,w}$. In this paper, we use the index $i = 1$, 2, ..., $n$ to denote draw $i$ from the forecast distribution, resulting in a collection of realizations $\tilde{y}_{r,s,w,i}$. In probabilistic settings we can no longer rely on the coherence definition given in Eq 3. Although various definitions for probabilistic coherence exist [18, 22], in this paper, we choose the intuitive presentation of Gamakumara et. al [22]. We say that the density $f(\tilde{\boldsymbol{y}}_{s,w})$ is *probabilistically coherent* if

$$f(\tilde{\boldsymbol{y}}_{s,w}) \quad = 0 \quad \text{when} \quad \tilde{y}_{11,s,w} \neq \sum_{r=1}^{10} \alpha_r \tilde{y}_{r,s,w}. \tag{16}$$

Here $f$ represents the joint density over all regions and the national level of the hierarchy. Note the close correspondence with the point forecast definition. However, probabilistic coherence does require specification of a joint distribution over regions. Intuitively, this definition says that any point in the support of $f$ that does not obey point-wise coherence is assigned zero probability, (i.e., has measure zero).

Probabilistic forecasts are binned into discrete distributions, where each bin represents a specific wILI level rounded to nearest first decimal. The FluSight challenge uses bins from {0.0, 0.1, 0.2, ..., 13.0, 100} to score discrete probabilistic forecasts. We score forecasts using both "single-bin skill" and "multibin-bin skill" [23, 24]. We do this to demonstrate the method's utility on the historical multi-bin skill, as well as the newly adopted single-bin skill. However, unlike the "log score" (the logarithm of forecast skill) used in the CDC FluSight challenge, we exponentiate the logarithm so that skills remain on the interpretable [0, 1] probability scale. Like log score, skill is a thresholding scoring rule, where a forecast is deemed correct if it is within a certain distance of the truth. For single-bin scoring, the probability assigned to the true target $Z_t$ (e.g., a one-week-ahead forecast) corresponding to region $r$ in flu season $s$ and epiweek $w$ under single-bin is computed as

$$p_{r,s,w,Z_t} \quad = \frac{1}{n} \sum_{i=1}^{n} \mathbf{1}(\tilde{y}_{r,s,w,i} \in Z_t), \tag{17}$$

where $Z_t$ is the true target.

We can extend this to multi-bin as

$$p_{r,s,w,Z_t} = \frac{1}{n}\sum_{i=1}^{n}\mathbf{1}(\tilde{y}_{r,s,w,i} \in B),\tag{18}$$

where $B = [Z_t - b, Z_t + b]$ for the true target $Z_t$ and some pre-defined threshold $b$. Using a thresholding evaluation metric breaks the guaranteed equal or improved performance of the coherent forecasts when evaluated with MSE.

To see why, consider again the example from Section 1 with $\mathbf{y}^T = [1, 1, 1]$. For $i = 1, 2, \ldots,$ $n = 10000$, we independently draw

$$\tilde{y}_{\text{HHS1},i} \sim \text{N}(0.5, 0.05),\tag{19}$$

$$\tilde{y}_{\text{HHS2},i} \sim \text{N}(0.5, 0.05),\tag{20}$$

$$\tilde{y}_{\text{Nat},i} \sim \text{N}(1, 0.05),\tag{21}$$

defining $\tilde{\mathbf{y}}_i^T = [\tilde{y}_{\text{HHS1},i}, \tilde{y}_{\text{HHS2},i}, \tilde{y}_{\text{Nat},i}]$ and $\hat{\mathbf{y}}_i^T = \mathbf{X}(\mathbf{X}^T\mathbf{X})^{-1}\mathbf{X}\tilde{\mathbf{y}}_i$. Fig 3 shows the $n$ draws of $\tilde{\mathbf{y}}$ and $\hat{\mathbf{y}}$. The single-bin skill counts all forecasts that equal the true value (rounded to the first decimal

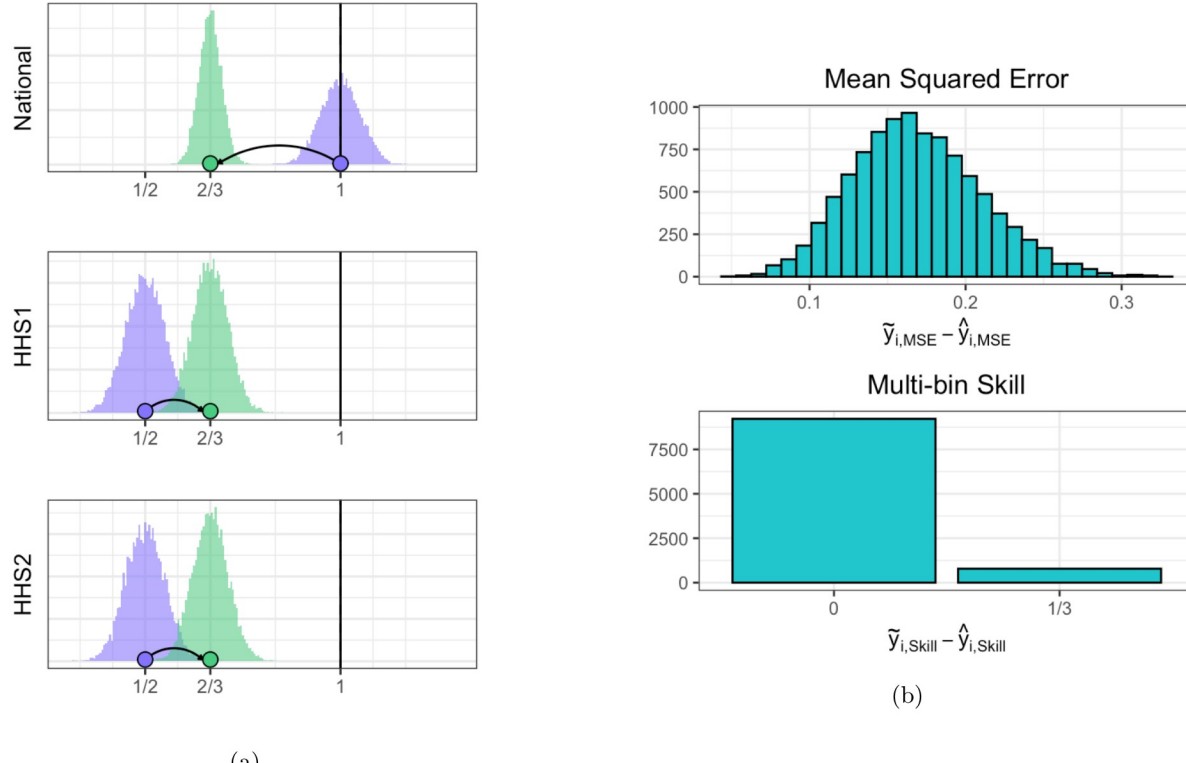

(a)

(b)

**Fig 3. Graphical example of how mean squared error (MSE) can decrease while skill gets worse for two region example. A:** Purple histograms represent the 10,000 realizations of $\tilde{\mathbf{y}}$, while green histograms are the corresponding $\hat{\mathbf{y}}$s. The purple and green points illustrate a particular example of the projection matrix forecasting process. The solid vertical lines denote the true value for each region. **B:** Top panel shows distribution of MSE for $\tilde{\mathbf{y}}$ minus corresponding MSE for $\hat{\mathbf{y}}$. MSE for $\tilde{\mathbf{y}}$ is greater than the MSE for $\hat{\mathbf{y}}$ for all realizations. **B:** Bottom panel shows single-bin skill score for $\tilde{\mathbf{y}}$ minus skill score for $\hat{\mathbf{y}}$. The incoherent $\tilde{\mathbf{y}}$ forecasts are better or equal to the skill for the coherent forecasts for all iterations, with an average improvement greater than 0. This shows the the MSE of the coherent forecasts has decreased (since the difference between the original and projected is positive) and the forecast skill has decreased (since the difference between the original and projected is again positive). Since a decrease in MSE means an improvement and a decrease in forecast skill means a lack of improvement, we see that coherence can have opposite effects on the two scores.

place) as correct. For every realization $i$, the MSE for $\hat{\boldsymbol{y}}_i$ is less than the MSE of $\tilde{\boldsymbol{y}}_i$. However, the single-bin skill for the coherent forecasts is 0 while the skill for incoherent forecasts is 0.32 and the skill for the incoherent forecasts is always better than or equal to the skill for the coherent forecasts. This is because all of the national forecasts that fell on top of the truth are projected out of the true bin while the incorrect regional forecasts are moved closer to the correct region, but not close enough to fall inside it. The result is a collection of coherent forecasts with better (lower) MSE, but also lower forecast skill than the incoherent forecasts.

## Problem statement

On one hand, we have a guarantee that the MSE of point forecasts projected into the data generating process space can get no worse under the OLS projection method. On the other, we have an explicit example of forecast skill decreasing as a result of forecasts projected into the data generating process space. This seeming inconsistency leads us to the central question of this paper: Can probabilistic forecast coherence be used to improve forecast skill when forecasting influenza? The remainder of this paper will investigate this question. In this analysis, we focus only on short-term (1-4 week ahead) targets. The definition of coherence is less clear on seasonal targets, since knowing the regional season peaks does not inform the national season peak.

## Methods

In order to investigate the question posed in Section 1.4 we developed four methods to sample from probabilistically coherent forecast densities (Fig 4). To begin, we define the joint density of original forecast densities, drawn independently for each region:

$$f(\tilde{\boldsymbol{y}}) = \prod_{r=1}^{11} f_r(\tilde{y}_r). \tag{22}$$

Previous approaches have factored $f(\tilde{\boldsymbol{y}})$ into a bottom-up density, where

$$f(\tilde{\boldsymbol{y}}) = \delta(\tilde{y}_{11}|\tilde{y}_{1:10})h(\tilde{y}_{1:10}|\theta), \tag{23}$$

where $\delta$ is a Dirac delta density centered at $\tilde{y}_{11} = \sum_{r=1}^{10} \alpha_r \tilde{y}_r$, $\theta$ is a parameter(s) estimated from training data, and $h$ is a joint density over all 10 regions [18].

The bottom-up model of Eq 23, while probabilistically coherent, lacks robustness in two key ways. First, it requires historical training data to estimate $\theta$. This is not always possible, particularly in emerging epidemic settings. Second, the bottom-up approach ignores information encoded in the original forecast density for the national region $\tilde{y}_{11}$. We instead develop methods that take draws from $f(\tilde{\boldsymbol{y}})$ of Eq 23 and produce draws for a probabilistically coherent $f^*(\hat{\boldsymbol{y}})$. These methods address both of the shortcomings of the bottom-up density of Eq 23: they do not discard national scale forecasts and they do not require training data.

In what follows we consider four projection approaches for sampling from the probabilistic coherent density, $f^*(\hat{\boldsymbol{y}})$, as well as two baseline approaches. We first consider the scenario where forecast distributions are assumed to be uncorrelated across regions and where forecasts from each region are weighted equally. This allows us to sample from the original forecast distribution $f(\tilde{\boldsymbol{y}})$ and apply a point-wise coherence projection matrix to each independently drawn sample $\tilde{y}_i$. In the second approach, we assume the geographical units are positively correlated and forecasts from each region are weighted equally. This correlation structure reflects our knowledge that during an epidemic, forecast models that tend to under predict wILI at the

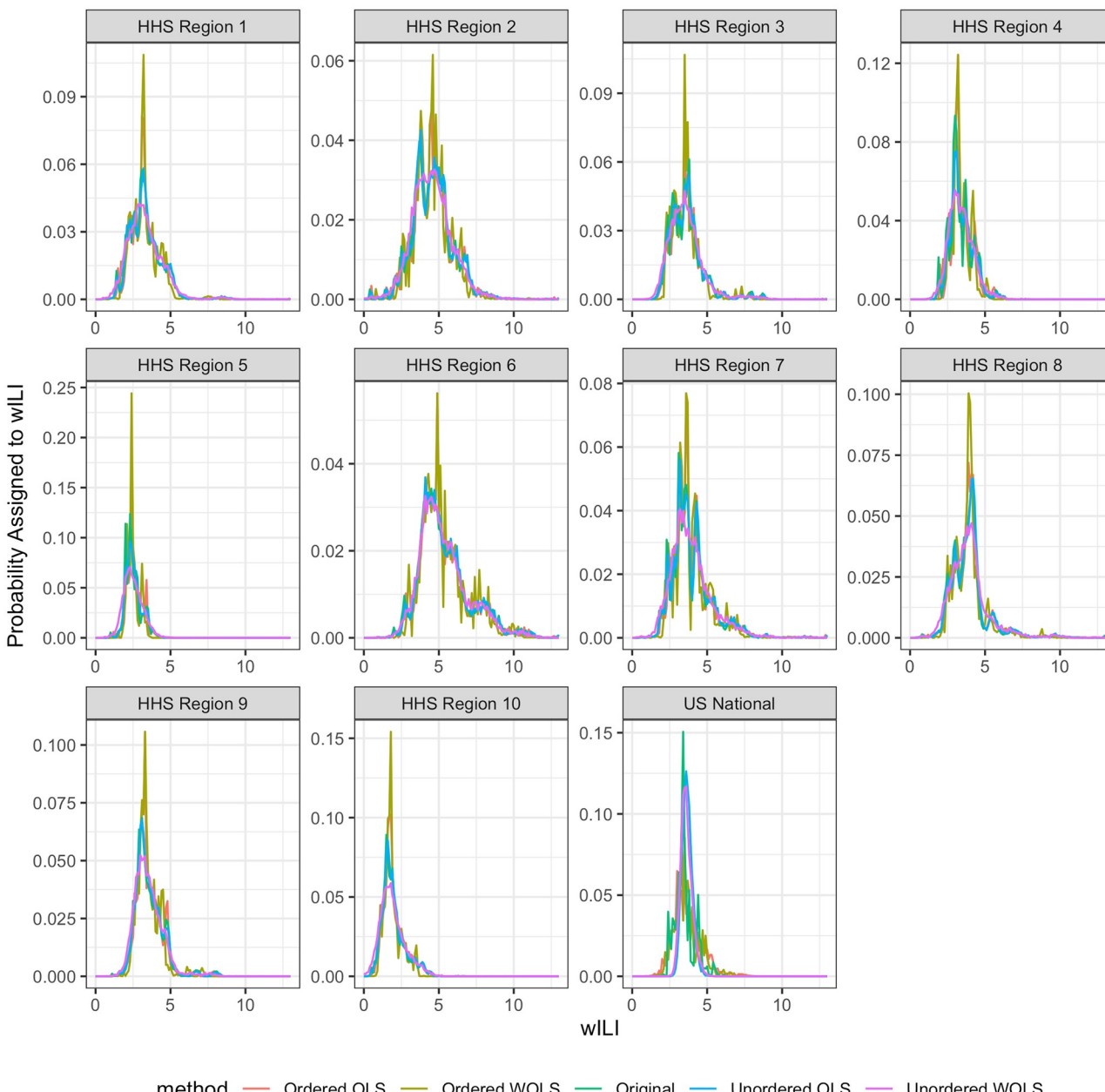

**Fig 4. Real data example of model predictive densities for the 1 week ahead target on epiweek 201901 for the 2018/2019 season across all 11 regions.** The y-axis represents the probability density for a given wILI bin value on the x-axis. Notice how the regional samples do not change much under the coherence constraint, but the national forecasts noticeably change. We can also see variable levels of density "smoothing" produced by each method, with the greatest amount of smoothing under the Unordered weighted ordinary least squares (WOLS) method. This smoothing of forecast density also lowers the magnitude of the peak density across all HHS regions, but increases the magnitude of the peak in the nation. However, the overall location of the forecast density remains consistent across all projection methods.

regional level, will also do so at the national level. The correlation induced creates a positive correlation between all the regions and the nation. Third, we consider the case where regions are uncorrelated but we allow forecasts for regions with larger populations to be weighted more heavily than those with smaller populations. This allows us weight forecasts made from regions with larger populations (i.e. larger sample sizes) more heavily in the projection. Fourth,

we consider forecasts as both correlated across regions and variable according to population of the region. Finally, we include the original forecast distribution and the bottom-up method as reference models.

## Ordinary least squares

Our first approach requires samples from the independent forecast distributions, defined as follows:

$$\tilde{y}_{1,i} \sim f_1(\tilde{y}_1), \quad \tilde{y}_{2,i} \sim f_2(\tilde{y}_2), \quad \cdots, \quad \tilde{y}_{11,i} \sim f_{11}(\tilde{y}_{11}), \tag{24}$$

where $i$ indexes samples and where we have dropped the season, epiweek, and model index for simplicity. We then apply the projection matrix to the column vector

$$\hat{\boldsymbol{y}}_i = \boldsymbol{P} \begin{pmatrix} \tilde{y}_{1,i} \\ \tilde{y}_{2,i} \\ \cdots \\ \tilde{y}_{11,i} \end{pmatrix} = \boldsymbol{P}\tilde{\boldsymbol{y}}_i. \tag{25}$$

This approach is graphically demonstrated in Fig 2. Using the projection matrix on each sample guarantees that the resulting empirical probability mass function satisfies the probabilistic coherence definition of Eq 16. Algorithm 1 (Fig 5) outlines how to produce samples from $f^*(\hat{\boldsymbol{y}})$. In practice, the CDC submission files specify probability distributions as binned probability mass functions. Independent forecasts $\tilde{y}_i$ are sampled from these binned probability mass functions.

1: **for** r = 1,2,...,11 **do**
2:     **for** i = 1,2,....,n **do**
3:         $\tilde{y}_{r,i} \sim f_r$
4:     **end for**
5: **end for**
6: **for** i = 1,2,....,n **do**

7:     $\hat{\boldsymbol{y}}_i = \boldsymbol{P}\tilde{\boldsymbol{y}}_i = \boldsymbol{P} \begin{pmatrix} \tilde{y}_{1,i} \\ \tilde{y}_{2,i} \\ \cdots \\ \tilde{y}_{11,i} \end{pmatrix}$

8: **end for**

**Fig 5. Unordered OLS sampling from probabilistically coherent joint distribution given a collection of marginal distributions.** Note that the corresponding weighted ordinary least squares (WOLS) method is obtained by replacing $\boldsymbol{P}$ with $\boldsymbol{P}_w$.

## Ordered ordinary least squares

The unordered OLS approach assumes no correlation between the error structures regionally and nationally. That is, each sample is generated independently. In the second approach we induce a correlation structure between the forecast distributions. We begin with the same set of samples as defined in Eq 22. However, before applying the projection matrix we first compute the order statistics. We then apply the projection matrix to the column vector of the aligned order statistics:

$$\hat{\boldsymbol{y}}_i = \boldsymbol{P} \begin{pmatrix} \tilde{y}_{1_{(i)}} \\ \tilde{y}_{2_{(i)}} \\ \ldots \\ \tilde{y}_{11_{(i)}} \end{pmatrix} = \boldsymbol{P}\tilde{\boldsymbol{y}}_{(i)}$$

where $\tilde{y}_{r_{(i)}}$ is the $i^{th}$ order statistic for the empirical distribution $f_r(\tilde{y}_r)$ for region $r$.

Both the ordered and unordered OLS approaches lead to empirical distributions that are probabilistically coherent, however, the ordered OLS approach induces a correlation structure where low regional wILI forecasts are tied to low national wILI forecasts and vice versa: similar to the Schaake Shuffle [25]. In practice, the ordered OLS algorithm amounts to first sorting the samples drawn independently at each region and then applying the projection matrix to the sorted samples as outlined in Algorithm 2 (Fig 6).

## Weighted ordinary least squares

In order to incorporate our uncertainty of the independent forecasts made in each region, we generalize the OLS method to weighted ordinary least squares (WOLS), where the weight matrix $\boldsymbol{V}$ is a diagonal matrix with entries corresponding to the inverse of the population

1: **for** r = 1,2,...,11 **do**
2:     **for** i = 1,2,....,n **do**
3:         $\tilde{y}_{r,i} \sim f_r$
4:     **end for**
5: **end for**
6: **for** i = 1,2,....,n **do**
7:     Set $y_{(i)}$ to the $i^{th}$ order statistics
8:     $\hat{\boldsymbol{y}}_i = \boldsymbol{P}\tilde{\boldsymbol{y}}_{(i)} = \boldsymbol{P} \begin{pmatrix} \tilde{y}_{1_{(i)}} \\ \tilde{y}_{2_{(i)}} \\ \ldots \\ \tilde{y}_{11_{(i)}} \end{pmatrix}$
9: **end for**

**Fig 6. Ordered OLS sampling from probabilistically coherent joint distribution given a collection of marginal distributions.** Note that the corresponding WOLS method is obtained by replacing $\boldsymbol{P}$ with $\boldsymbol{P}_V$.

weights for the region,

$$diag(\boldsymbol{V}^{-1}) = \{\alpha_{\text{HHS1}}, \alpha_{\text{HHS2}}, \ldots, \alpha_{\text{HHS10}}, 1\} \tag{26}$$

where $\alpha_j$ is the normalized population weight defined in Section 1.2. The projection $\boldsymbol{P_V}$ matrix becomes,

$$\boldsymbol{P}_V = \boldsymbol{X}(\boldsymbol{XVX}^T)^{-1}\boldsymbol{X}^T\boldsymbol{V} \tag{27}$$

The WOLS maintains a coherent projection in that applying $\boldsymbol{P_V}$ to a vector projects the vector into the column space of $\boldsymbol{X}$, but allows us to treat each forecast with a different degree of certainty. The weighted projection can be extended to allow for correlation using the ordered approach described in Section 2.2, but replacing $\boldsymbol{P}$ with $\boldsymbol{P_V}$.

## Ordered weighted ordinary least squares

Finally, we can apply the same ordering step to the weighted least squares projection. This requires computing the order statistics and applying the weighted projection matrix ($\boldsymbol{P_V}$)

$$\hat{\boldsymbol{y}}_i = \boldsymbol{P}_V \begin{pmatrix} \tilde{y}_{1_{(i)}} \\ \tilde{y}_{2_{(i)}} \\ \ldots \\ \tilde{y}_{11_{(i)}} \end{pmatrix} = \boldsymbol{P}_V \tilde{\boldsymbol{y}}_{(i)}$$

This leads to a set of forecasts that are both correlated and weighted by a normalized population measure to reflect our uncertainty of the independent forecasts by region. This method is a composition of Section 2.2 and 2.3.

## Experimental setup

In order to examine the effects of the unordered and ordered OLS/WOLS approaches on forecast skill, we use submission files for the 2016/2017, 2017/2018, and 2018/2019 seasons that have been submitted to the CDC and are uploaded to the central repository [17]. For each evaluation season, we obtain a list of all models submitted, and evaluate all four approaches across both single and multi-bin scoring for 1-4 week ahead targets across epiweeks 44-17. Any model that did not have a complete set of submission files for all 1-4 week ahead targets, all epiweeks, and all regions was discarded. The sample sizes for the evaluation are included in Table 1, where we define an evaluation point as a unique region, season, model, epiweek, and target combination. As we can see from Table 1, the sample sizes are quite substantial when aggregating over evaluation points. We use the 2010 U.S. Census weights across all seasons as an estimate of $\alpha_r$ for each region [21]. The correlation between the weighted combination of the regionally reported wILI by the CDC and the nationally reported wILI is >.99. Using the 2010 U.S. Census weights is a reasonable approximation to the weights used by the CDC.

**Table 1. Experimental setup for evaluating probabilistic coherence approaches.** An evaluation point is defined as a unique region, season, epiweek, target, and model combination.

| Season | Number of Models | Number of Evaluation Points |
|---|---|---|
| 2016/2017 | 24 | 27,456 |
| 2017/2018 | 24 | 27,456 |
| 2018/2019 | 35 | 40,040 |

**Table 2. Percent of forecasts improved over original forecast distribution for the four coherence methods described, in addition to the baseline bottom up model under both single-bin and multi-bin skill.** We omit the independent forecasts since they are the reference model for percentage of forecasts improved. Notice that WOLS unordered showed the greatest improvement under single-bin scoring but showed the least improvement under multi-bin scoring. This demonstrates that the scoring rule used influences the performance of the coherence methods.

| Percent of forecasts improved | Single-Bin | Multi-Bin |
| --- | --- | --- |
| Unordered OLS | 64 | 65 |
| Ordered OLS | 51 | 90 |
| Unordered WOLS | 79 | 53 |
| Ordered WOLS | 31 | 57 |
| Bottom Up | 39 | 59 |

## Results

In what follows we consider the combination of a model and a season as the fundamental unit of analysis on which to base our conclusions. As a forecaster, the main question under consideration is whether applying forecast coherence will improve the average forecast skill of a given model in an upcoming season. As we can see from Table 2 the results vary heavily by scoring method used. For the main results, we use single-bin skill as this is the most up to date skill used in the FluSight challenge, but we also present various summary results for multi-bin to highlight the differences in results under different scoring mechanisms.

Under single-bin scoring we saw 79% of models improve under the unordered WOLS method, 64% of models improve with unordered OLS, 51% of models improve with ordered OLS, and 31% of models improve with ordered WOLS. As seen in Fig 7, the increase in skill ranged from -.005 to.025 under single-bin with a mean of.002 and variance 1.7e-05. This shows that the magnitude of improvement is greater than the magnitude of decrease. This asymmetry is even more pronounced in the multi-bin scores (Fig 7 right), with the greatest improvement in skill of.15 and biggest loss in skill of -.005. The average increase under multi-bin scoring for the ordered OLS method was.0024 and variance 2.8e-03.

We can see that 17 model season combinations got worse under unordered WOLS and single-bin scoring and only 5 models got worse under ordered OLS and multi-bin scoring (Fig 7). This suggests that modeling the correlation structure has a significant effect on the results under single-bin scoring, with unordered WOLS improving significantly more than ordered WOLS (79% vs 31%). However, the correlation structure did seem to improve skill under multi-bin scoring, with ordered OLS improving significantly more than unordered OLS (90% vs 51%). Further work that allows for assumptions in between these two extremes is required. The results also suggest that the weighting is important under single-bin scoring.

The results are consistent across targets (see Fig 8 left), where a majority of model/season forecast skill improves over the forecast skill of the independent forecasts under the unordered WOLS method. The median increase in forecast skill under the unordered WOLS method was.0039 (variance 2.52e-05) at 1-week ahead,.0024 (variance 1.19e-05) at 2-week ahead,.0021 (variance 8.44e-06) at 3-week ahead and.0020 (variance 8.27e-06) at 4-week ahead. We can see that the difference in forecast skill diminishes as horizon increases. This suggests that coherence has the greatest benefit at shorter time horizons, where forecasts are more accurate. The other 3 projection methods do not show consistent improvement across targets.

We also break down the results by season (Fig 8 right). Here we see that the unordered WOLS method consistently improves forecast skill, even under the 2017-2018 epidemic year. In the 2016-2017 season all but 4 models improved under the unordered WOLS method (Fig 7 left), in 2017-2018 all but 4 models improved and in 2018-2019 all but 8 models improved.

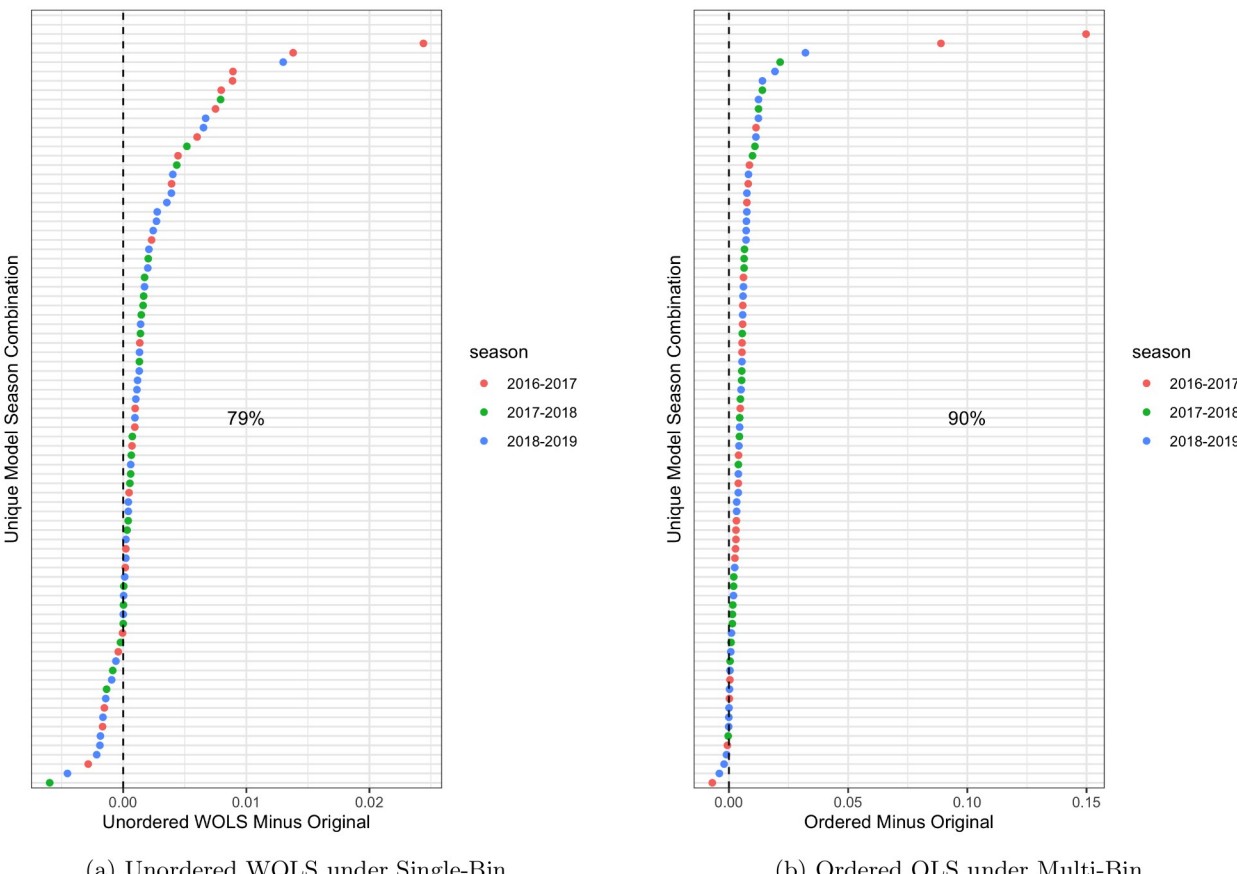

(a) Unordered WOLS under Single-Bin          (b) Ordered OLS under Multi-Bin

**Fig 7. Best performing method under single-bin (left) and mutli-bin (right) in terms of forecast skill averaged over all targets (1-4 week ahead), regions (HHS1-10 & National) and broken down by model-season combination.** The y-axis represents a unique season model combination which has been made anonymous to protect participant teams identity.

The magnitude of improvement does vary significantly by season. The median improvement in 2016-2017 was.0042 (variance 2.24e-05), in 2017-2018 was.0015 (variance 4.96e-06) and in 2018-2019 was.002 (variance 5.30e-06). The other three projection methods do not show consistent improvement across seasons.

Finally, the results are consistent across HHS regions (Fig 8 bottom) with the unordered WOLS method improving all HHS regions forecast skill. However, the forecast skill does not significantly change in the national region. The median improvement in HHS regions (excluding national) was 0.0028 (variance of 1.71e-05). The median decrease in the national level was -0.0008 (variance 5.59e-05). We explore this result in the Discussion Section. The other three projection methods do not show consistent improvement across regions.

## Discussion

Forecast coherence is a simple tool to improve forecast skill of short-term predictions in systems with hierarchical structures. In order to demonstrate this, we first defined probabilistic coherence, and showed that the results in the literature surrounding point forecast coherence do not naively transfer over to probabilistic forecasting. Guarantees for improvements in MSE do not directly transfer over to the CDC FluSight forecast skill metric of probabilistic forecast performance. However, by leveraging the definition of probabilistic coherence, we were able

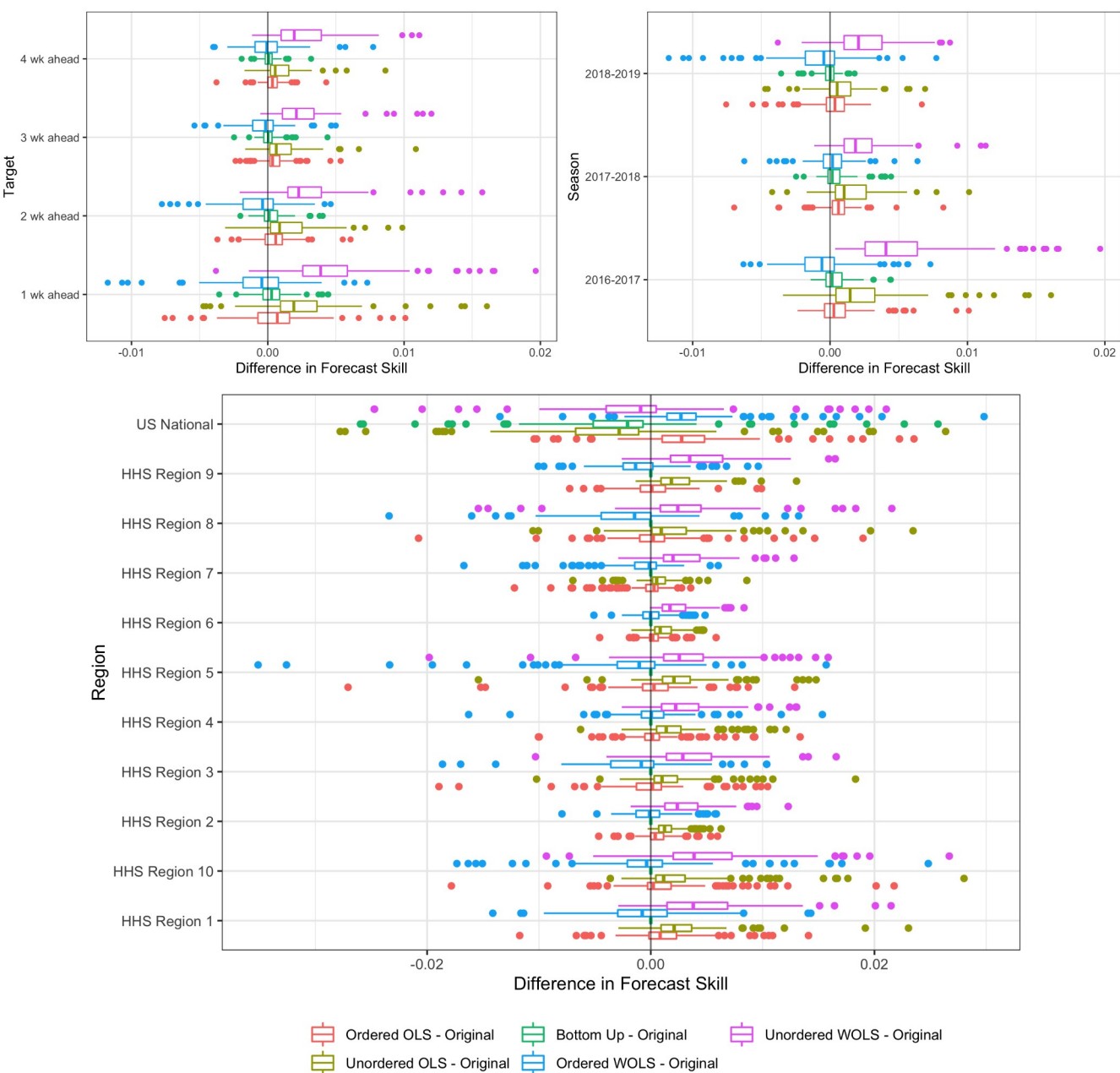

**Fig 8. Difference between single-bin forecast skill of projection method and forecast skill of independent forecasts averaged over all regions and epiweeks broken down by target (left), season (right), and region (bottom).** Each point represents a single model-season combination. Box-whisker forecasts and represent the inter-quartile range as well as the maximum and minimum in forecast skill difference between projected method and independent forecasts. The improvements in single-bin forecast skill are consistent across season and target for the unordered WOLS. However, the improvements are only consistent across the HHS regions, not the national region.

to generate coherent samples by first sampling from a collection of independent forecast distributions over all regions and projecting them onto a coherent subspace. This projection method is generic and allows for both correlated and uncorrelated projections and weighted and unweighted projections. By exploiting the underlying variability of the forecasts using normalized population size as a proxy for forecast variance, we were able to improve average forecast skill when broken down by model, season and target.

In practice, the unordered and ordered OLS/WOLS methods are very appealing due to the lack of training data required and the operational simplicity of manipulating a submitted forecast, without requiring adjustments to the model code itself. No knowledge of the process model used to generate forecasts is required, only the resulting predictive density. Even though the benefits in forecast skill are small in magnitude, there is little cost to implementing coherence in practice and, especially for the unordered WOLS method, the frequency of forecasts improved is high (79%).

Our experiments lead to the following conclusions:

- **Forecast coherence can benefit forecast skill, but the average benefits are small**. Using the unordered WOLS method, we can improve short-term forecast skill with high likelihood (79% of model/seasons). We see a small improvement but with little to no cost in terms of parameter estimation and implementation difficulty. This makes the unordered WOLS method a clear choice to use when submitting forecasts to the CDC FluSight challenge. These benefits are consistent across region, season, and target breakdowns. While the improvements are modest, the .002 average increase in forecast skill is significant enough to change model rankings. In particular, in the 2018-2019 season, an average increase would have moved 3 out of 33 models up a ranking [26]. However, some models do get worse under the unordered WOLS method, suggesting that forecasters implement coherence and perform interval cross-validation to decide.

- **Weighting the forecasts by the variance of the region improves scores**. Under single-bin scoring, we see the biggest improvement when we take the variability of the forecasts by region into account. We do this by weighting the forecasts by the inverse of the population size of each region (where the nation receives weight 1). Region population size serves as a reasonable proxy for the underlying forecast variability, without relying on historical forecast data to estimate region variances. Regions with larger populations should in principal have more reliable forecasts. Using population size as an estimate of the variance, clearly demonstrates a benefit over the unordered OLS method which weights all forecast distributions equally. However, improvements may increase by weighting model specific region variances in the projection matrix.

- **Coherence alters the variance of the forecasts** As we can see from Fig 9 the projection methods all change the average forecast variance when broken down by region. Under probabilistic scoring, this causes significant changes in the forecast skill of the methods. The optimal method under single-bin scoring is the unordered WOLS method. We can see from both Figs 4 and 9 that the variance increases and the distribution is smoothed. This is a desirable property under single-bin scoring, where over-confident forecasts are penalized disproportionately (due to the asymmetry of the logarithm). However, we also see that the variance of the unordered WOLS method is reduced in the National region. This corresponds with a lower average skill increase in the National region under the unordered WOLS method as seen in Fig 8.

- **The naive bottom-up method does not perform as well as projection methods.** The bottom-up method is an intuitive coherence strategy with minimal implementation effort which ignores the independent national level forecast. Although it is appealing due to its simplicity, we were able to significantly improve forecast skill under both single-bin and mutli-bin scoring by using projection methods.

- **Some models get worse under coherence methods.** For the 17 models that got worse under unordered WOLS method and single-bin scoring, the average variance of the forecast distribution across all models, targets, locations, and seasons was 2.36, whereas the average

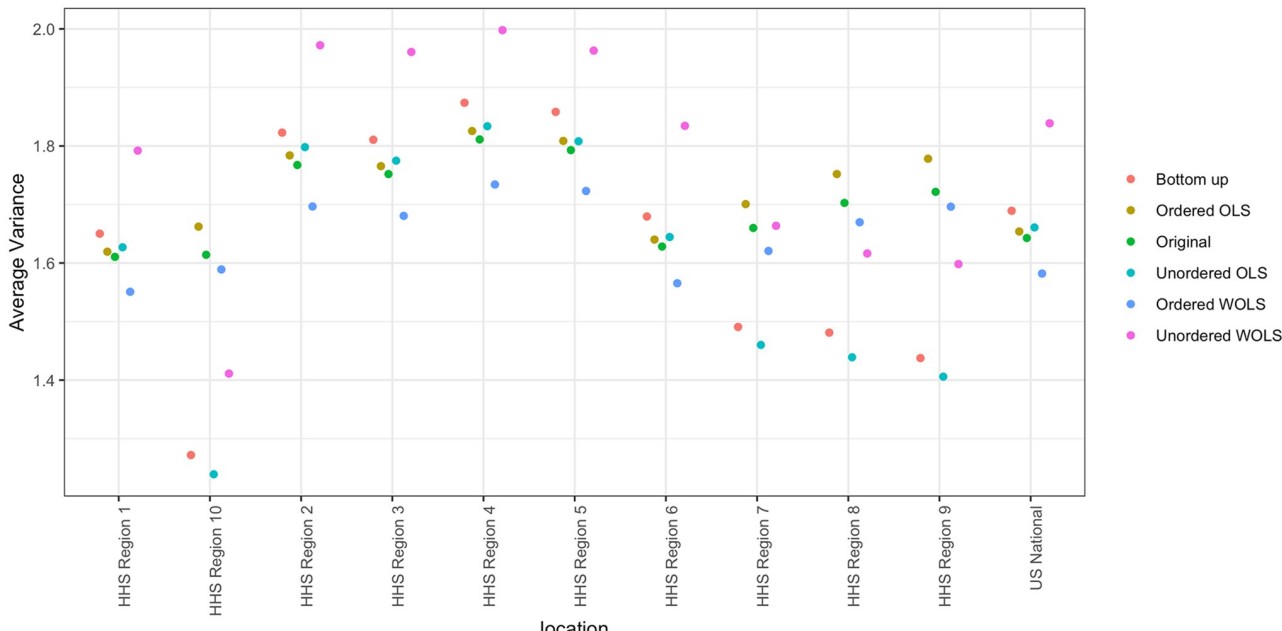

**Fig 9. Average variance of forecasts, averaged over season, epiweek, target, and model.** Notice that the unordered WOLS increases the variance across HHS regions, which is reflected in the improvements under single-bin scoring. However, the variance of the unordered WOLS decreases at the national level, which is also the only region without significant benefit under single-bin scoring. The optimal model under multi-bin scoring (ordered OLS) retains the same variance of the original forecast distribution for the HHS regions, but slightly increases the variance slightly for the nation. This demonstrates the effect of the scoring has on projection method choice.

variance of the forecast distribution across all models, targets, locations, and seasons was 1.77. This suggests that models that already are highly variable may not improve. However, we would require more detailed model information to completely ascertain why some models decreased in skill and others improved.

- **The choice of scoring rule matters**. We can see that the optimal method for single-bin scoring is not the optimal method for multi-bin scoring. The optimal projection method for single-bin scoring is unordered WOLS, which only showed 53% of forecasts improving under multi-bin, the lowest of any other methods. The optimal projection method under multi-bin scoring is ordered OLS, which only showed a 51%, the third lowest of any other method. This suggests that single-bin and multi-bin scoring are capturing different features of the forecast distribution. As seen in Fig 9, the unordered WOLS method increased the variance of the forecast distribution relative to the original forecasts. However, the ordered OLS method only slightly increased the variance of the forecast distribution relative to the original forecasts. This suggests that widening the variance under single-bin scoring increases forecast skill on average, which is consistent with single-bin scoring only counting probability density that falls exactly over the bin containing the truth. However, recent research has shown that multi-bin scoring is an approximation to common proper scoring rules such as the continuous rank probability score [27]. This suggests that in the broader probabilistic forecasting realm, the results under the multi-bin scoring rule might be more applicable.

Although projection methods are simple to implement and result in small but significant changes in forecast skill, there is still significant room for improvement. Recent work by Taieb et. al has explored copula based techniques to combine the independent forecast distributions

from the regional level into a joint distribution with a specified covariance structure [18]. Wickramasuriya et. al have also explored various projections using a weighted least squares method [19]. This weight matrix also represents the correlation structure between the independent forecasts but can be estimated from historical forecast accuracy. It is clear that, unlike in the point forecast setting, exploring various correlation structures in the probabilistic setting has a drastic effect on the results. Given historical training data, one could estimate the error correlation specific to a given process model which could potentially lead to an even greater increase in forecast skill. However, in the absence of historical training data, forecasters can still leverage coherence to improve probabilistic forecast skill. In all, these results suggest that simple and fast methods can improve probabilistic forecasts of systems where the available data has a natural hierarchy. In practice, we recommend using cross-validation to choose the appropriate method for an individual forecasting model. While the results suggest that unordered WOLS is the correct choice, Fig 7 shows that some models may decrease in skill, so internal validation is advised. Using the example of seasonal influenza forecasts in the US, we show that enforcing coherence provides a high likelihood of improvement in forecast accuracy, and in general may provide opportunities for improvement in forecast accuracy in this and other real-world application settings.

## Supporting information

**S1 Appendix. Proof of MSE improvement under coherence.**
(PDF)

## Acknowledgments

The authors thank C.C. Essix for her encouragement and support of this work, as well as the helpful conversations with Dr. Brian Weaver. The content is solely the responsibility of the authors and does not necessarily represent the official views of CDC, NIGMS or the National Institutes of Health.

## Author Contributions

**Conceptualization:** Graham Casey Gibson, Dave Osthus.

**Data curation:** Graham Casey Gibson, Dave Osthus.

**Formal analysis:** Graham Casey Gibson, Dave Osthus.

**Funding acquisition:** Nicholas G. Reich, Dave Osthus.

**Investigation:** Graham Casey Gibson, Dave Osthus.

**Methodology:** Graham Casey Gibson, Kelly R. Moran, Nicholas G. Reich, Dave Osthus.

**Project administration:** Dave Osthus.

**Resources:** Dave Osthus.

**Software:** Graham Casey Gibson, Dave Osthus.

**Supervision:** Dave Osthus.

**Validation:** Graham Casey Gibson.

**Visualization:** Graham Casey Gibson, Kelly R. Moran, Dave Osthus.

**Writing – original draft:** Graham Casey Gibson, Dave Osthus.

**Writing – review & editing:** Graham Casey Gibson, Kelly R. Moran, Nicholas G. Reich, Dave Osthus.

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
