## [Decision Letter · Decision Letter 0]

12 Feb 2020

Dear Mr. Gibson,

Thank you very much for submitting your manuscript "Improving Probabilistic Infectious Disease Forecasting Through Coherence" for consideration at PLOS Computational Biology. As with all papers reviewed by the journal, your manuscript was reviewed by members of the editorial board and by several independent reviewers. The reviewers appreciated the attention to an important topic. Based on the reviews, we are likely to accept this manuscript for publication, providing that you modify the manuscript according to the review recommendations.

Sincerely,

Benjamin Althouse

Associate Editor

PLOS Computational Biology

Virginia Pitzer

Deputy Editor

PLOS Computational Biology

[LINK]

Reviewer's Responses to Questions

**Comments to the Authors:**

Reviewer #1: The authors consider a pertinent and useful problem of making probabilistic forecasting consistent across spatial hierarchies. They build upon recent methods for constructing coherent probabilistic forecasts and test its utility on CDC FluSight challenge's data produced by multiple modeling teams. As the authors note, this method generalizes to any system where forecasts have a geographical hierarchy or obey a similar known constraint.

Overall the paper is well written and includes sufficient graphical representations and examples to guide the reader. I am surprised that a simple method to enforce coherence improves overall forecast performance for most methods. I think this will be a useful post-processing tool for modelers in the FluSight challenge.

The authors mention they use methods inspired from Gamakumara et al. and Clark et al. While this seems to be the first application of forecast coherence in the context of FluSight challenge, the methodological novelty in this paper w.r.t. the problem itself is not clear. Would be good to highlight this through a related work comparison. The authors also state in the Introduction that 'demonstrated benefits of coherence in the point prediction setting do not necessarily translate to the probabilistic realm' ([19] in the submission). [18] seems to talk about coherence for probabilistic forecasting. More discussion is needed.

Is the positive correlation assumed by Ordered OLS only between region and national performance, or even among neighboring regions? Figure 7 does show the rationale between national and regional forecasts. In reality, I believe there is a case where the correlation structure is unknown (especially among regions given the underlying strain heterogeneity), and hence ordering forecasts across regions may not be straightforward. This may not be even learn-able from historical data, given the flu seasons have significant spatial variations across years.

In discussion, would be good for the authors to comment on preserving short-term and seasonal forecasts (onset/peak) consistency within each method and in the ensemble when altering each of these forecasts for hierarchical consistency. There is also the issue of consistency among 4 short-term forecasts. These are less straightforward to define, but some discussion will help.

Minor comments:

- While the authors state that a 'bottom up' method ignores national forecasts, it would still serve as a valid baseline for comparison. Are national forecasts always better than regional ones? It's not clear at the outset, given national has higher N but captures the spatiotemporal evolution of flu season only in a coarse sense.

- The paper seems to use 2010 Census numbers. Does CDC release 'official' weights (i.e., which year's census population estimates to use) for states/regions each influenza seasons? Though the authors report overall correlation, I assume using inexact weights may have impacts on individual forecasts (esp. in single-bin skill).

- The authors use multi-bin skill for evaluating the forecasts. Does this also translate to improved performance by the single-bin proper scoring rule that CDC is adopting starting 2019-20 season?

- References [18] and [20] seem to be duplicates (including page numbers).

- In Figure (4) use a darker black for true wILI.

- In Section 2.1, the graphical demonstration of Unordered OLS is cited as Figure 4. Is this correct?

- In Section 2.3, would be good to explicitly state 'evaluate both approaches on short-term forecasts across epiweeks 44-17'

- Typo in first paragraph of Section 3. The 'unordered OLS method' saw an improvement in two-thirds...

- Some explanation for the structure of the projection matrix (eqn (5)) will be useful.

- What is the y-axis in Figure 5? Is it each epiweek in the three test seasons?

Reviewer #2: The authors present a method for imposing appropriate hierarchy on independently generated, probabilistic, regional forecasts. The analysis performed on a wide variety of influenza forecasts suggests that the ordered ordinary least squares method will generally provide improvement to forecast skill. The authors provide an objective analysis of likelihood and quantity of improvement. The method(s) as described are straightforward to apply in this setting as well as other probabilistic forecasting settings. As such, I think this makes a nice contribution to the field. Very nice work.

Major Issues:

none

Minor Issues:

(By Section)

(Introduction)

"As part of this challenge, forecasters supply probabilistic forecasts for short-term and seasonal targets at both the national and regional levels corresponding to weighted influenza-like illness (wILI), which measures the proportion of outpatient doctor visits at reporting health care facilities where the patient had an influenza-like illness (ILI), weighted by state population."

- sentence is difficult to understand and 'had an influenza-like illness' is misleading -> exhibits influenza like symptoms

"directly computed using state population weighted ILI"

- this sounds like weighted ILI at the state level rather than the population-weighted average of state ILI

"The CDC estimates ILI as the ratio of patients presenting with a cough and fever equal to or above 100° Fahrenheit over the total number of patients presenting at health care providers [5]"

- It is my understanding that ILI is fever AND (cough OR sore throat). Not sure this is the best reference. Maybe something from ILINet?

- Might be worth providing a little more detail about the FluSight forecasts for the uninitiated reader. Forecast targets, weeks available, etc.

(1.1)

"The CDC reports the wILI data using epidemic weeks, called epiweeks, instead of calendar weeks [21]."

- I wasn't able to find this source. add url or use something from MMWR

(1.3)

"We score the probabilistic forecasts using multi-bin skill, rather than multi-bin log score as used in the FluSight challenge......"

- Someone unfamiliar with the challenge may have trouble understanding this paragraph. Maybe start by introducing the set of bins Z, by which FluSight probabilistic forecasts are submitted. Then summarize skill and log score before continuing with the paragraph.

(2)

"Previous approaches have factored...[18]"

- Text implies there should be more than one reference

- Figure 4: vertical line for true wILI is difficult to distinguish from grid.

(2.2)

- Algorithm 2 might be more clear if it explicitly includes a step for sorting along the i index. Otherwise the reader must catch a minor change of notation to see the difference from Algorithm 1.

(3)

- Figure 5: axis tick marks are too small. Caption has typos

- Figure 6: specify box and whisker extents/limits

"In fact, the breakdown obscures any improvement at all, a consequence of using forecast skill as the primary metric, which is a geometric mean of log score."

- Log score has not been defined. It is not self-evident that this comment is true.

- Figure 7: what are the error units? wILI? weeks? both?

General comments:

- Manuscript still has some typos (see Figure 5 caption).

**Have all data underlying the figures and results presented in the manuscript been provided?**

Reviewer #1: Yes

Reviewer #2: Yes

PLOS authors have the option to publish the peer review history of their article (what does this mean?). If published, this will include your full peer review and any attached files.

Reviewer #1: No

Reviewer #2: No
---

## [Decision Letter · Decision Letter 1]

20 Aug 2020

Dear Mr. Gibson,

Thank you very much for submitting your manuscript "Improving Probabilistic Infectious Disease Forecasting Through Coherence" for consideration at PLOS Computational Biology. As with all papers reviewed by the journal, your manuscript was reviewed by members of the editorial board and by several independent reviewers. The reviewers appreciated the attention to an important topic. Based on the reviews, we are likely to accept this manuscript for publication, providing that you modify the manuscript according to the review recommendations.

Please address the minor points by reviewer 2.

Sincerely,

Benjamin Althouse

Associate Editor

PLOS Computational Biology

Virginia Pitzer

Deputy Editor

PLOS Computational Biology

[LINK]

Please address the minor points by reviewer 2.

Reviewer's Responses to Questions

**Comments to the Authors:**

Reviewer #1: I appreciate the revision the authors have done to the manuscript. In addition to increasing the readability, the methodology has also been revised to accommodate the single-bin scoring rules and comparisons with the bottom-up baseline. Additional discussion points are also very helpful in highlighting the novelty and the caveats (i.e., some models do not improve under coherence).

I feel my concerns with the earlier version have been adequately addressed and do not have any further comments on the paper.

Reviewer #2: Minor Comments:

- The acronym 'WOLS' appears in captions prior to being defined.

- Table 2 caption and text reference does not make clear that "percent increase" refers to "percent of forecasts improved". Globally, consider replacing statements like 'XX% improvement' with 'XX% of forecasts improved'

- Discussion, first bullet: "However, the some models..." should probably be "However, some models..."

- Discussion, fifth bullet: "Under the 17 models" might sound better "For the 17 models"

**Have all data underlying the figures and results presented in the manuscript been provided?**

Reviewer #1: Yes

Reviewer #2: Yes

PLOS authors have the option to publish the peer review history of their article (what does this mean?). If published, this will include your full peer review and any attached files.

Reviewer #1: No

Reviewer #2: No
---

## [Editor Report · Decision Letter 2]

14 Sep 2020

Dear Mr. Gibson,

We are pleased to inform you that your manuscript 'Improving Probabilistic Infectious Disease Forecasting Through Coherence' has been provisionally accepted for publication in PLOS Computational Biology.

Best regards,

Benjamin Althouse

Associate Editor

PLOS Computational Biology

Virginia Pitzer

Deputy Editor

PLOS Computational Biology

---

## [Editor Report · Acceptance letter]

17 Dec 2020

PCOMPBIOL-D-19-02203R2 

Improving Probabilistic Infectious Disease Forecasting Through Coherence

Dear Dr Gibson,

I am pleased to inform you that your manuscript has been formally accepted for publication in PLOS Computational Biology. Your manuscript is now with our production department and you will be notified of the publication date in due course.

With kind regards,

Livia Horvath
